# Complex Diagnostic Challenges in Glioblastoma: The Role of ^18^F-FDOPA PET Imaging

**DOI:** 10.3390/ph17091215

**Published:** 2024-09-15

**Authors:** David Sipos, Zsanett Debreczeni-Máté, Zsombor Ritter, Omar Freihat, Mihály Simon, Árpád Kovács

**Affiliations:** 1Department of Medical Imaging, Faculty of Health Sciences, University of Pécs, 7621 Pécs, Hungary; 2Doctoral School of Health Sciences, Faculty of Health Sciences, University of Pécs, 7621 Pécs, Hungary; 3Dr. József Baka Diagnostic, Radiation Oncology, Research and Teaching Center, “Moritz Kaposi” Teaching Hospital, Guba Sándor Street 40, 7400 Kaposvár, Hungary; 4Department of Medical Imaging, Medical School, University of Pécs, 7621 Pécs, Hungary; 5Department of Public Health, College of Health Science, Abu Dhabi University, Abu Dhabi P.O. Box 59911, United Arab Emirates; 6Department of Oncoradiology, Faculty of Medicine, University of Debrecen, 4032 Debrecen, Hungary

**Keywords:** glioblastoma, diagnostic imaging, MRI, PET, ^18^F-FDOPA

## Abstract

Glioblastoma multiforme (GBM) remains one of the most aggressive and lethal forms of brain cancer, characterized by rapid proliferation and diffuse infiltration into the surrounding brain tissues. Despite advancements in therapeutic approaches, the prognosis for GBM patients is poor, with median survival times rarely exceeding 15 months post-diagnosis. An accurate diagnosis, treatment planning, and monitoring are crucial for improving patient outcomes. Core imaging modalities such as Computed Tomography (CT) and Magnetic Resonance Imaging (MRI) are indispensable in the initial diagnosis and ongoing management of GBM. Histopathology remains the gold standard for definitive diagnoses, guiding treatment by providing molecular and genetic insights into the tumor. Advanced imaging modalities, particularly positron emission tomography (PET), play a pivotal role in the management of GBM. Among these, 3,4-dihydroxy-6-[^18^F]-fluoro-L-phenylalanine (^18^F-FDOPA) PET has emerged as a powerful tool due to its superior specificity and sensitivity in detecting GBM and monitoring treatment responses. This introduction provides a comprehensive overview of the multifaceted role of ^18^F-FDOPA PET in GBM, covering its diagnostic accuracy, potential as a biomarker, integration into clinical workflows, impact on patient outcomes, technological and methodological advancements, comparative effectiveness with other PET tracers, and its cost-effectiveness in clinical practice. Through these perspectives, we aim to underscore the significant contributions of ^18^F-FDOPA PET to the evolving landscape of GBM management and its potential to enhance both clinical and economic outcomes for patients afflicted with this formidable disease.

## 1. Introduction

Glioblastoma multiforme (GBM) is the most aggressive and lethal form of primary brain tumor, categorized as a grade IV astrocytoma by the World Health Organization (WHO). Representing approximately 15% of all intracranial neoplasms and 60–70% of all astrocytic tumors, GBM is characterized by rapid growth and a highly infiltrative nature, which complicates therapeutic interventions [1]. Despite advances in surgical techniques, radiotherapy, and chemotherapy, the median survival time for patients with GBM remains dismally low, typically ranging from 12 to 15 months post-diagnosis [1,2].

Molecularly, GBM is a heterogeneous disease marked by a multitude of genetic and epigenetic abnormalities. Common genetic alterations include mutations in the tumor suppressor genes TP53 and PTEN, the amplification of the EGFR gene, and deletions in the CDKN2A locus. Additionally, GBM tumors frequently exhibit aberrant signaling pathways such as the PI3K/AKT/mTOR and RAS/RAF/MEK/ERK pathways, contributing to uncontrolled cell proliferation, the evasion of apoptosis, and enhanced migratory and invasive capacities [3,4].

GBM presents significant challenges in both diagnosis and treatment due to the formidable barriers imposed by the blood–brain barrier (BBB) and the blood–brain tumor barrier (BTB) [1,2]. The BBB, a highly selective permeability barrier, restricts the passage of most therapeutic agents to maintain central nervous system (CNS) homeostasis, thus limiting the efficacy of chemotherapeutic drugs. In addition to the BBB, the BTB, which forms within the tumor microenvironment, exhibits heterogeneous permeability that can vary significantly between different regions of the tumor, further complicating drug delivery [5]. These barriers hinder the penetration and uniform distribution of diagnostic imaging agents and therapeutic compounds, making early detection and effective treatment difficult. Moreover, the invasive nature of GBM, characterized by diffuse infiltration into the surrounding brain tissue, complicates surgical resection and further limits the effectiveness of localized therapies [5,6]. Consequently, overcoming these barriers remains a critical area of research, necessitating the development of novel strategies such as nanoparticle-based drug delivery systems, BBB disruption techniques, and targeted molecular therapies to enhance diagnostic accuracy and therapeutic efficacy against GBM [5,6,7].

The current standard of care for GBM, known as the Stupp protocol, involves maximal safe surgical resection followed by concurrent radiotherapy and temozolomide (TMZ) chemotherapy, with subsequent adjuvant TMZ. However, the invasive nature of GBM often precludes complete surgical resection, and the tumor’s inherent resistance to radiation and chemotherapeutic agents further diminishes treatment efficacy [8]. Emerging therapeutic strategies are increasingly focusing on targeting the molecular underpinnings of GBM, including the development of novel targeted therapies, immunotherapies, and personalized medicine approaches aimed at improving patient outcomes [9].

Recent advancements in genomic and transcriptomic profiling have provided deeper insights into the molecular landscape of GBM, revealing potential biomarkers for diagnosis, prognosis, and therapeutic targeting. Despite these advancements, the complexity and adaptability of GBM necessitate continued research into its molecular mechanisms and the development of innovative treatment modalities [9,10].

The main goal of this article is to discuss the role of Magnetic Resonance Imaging to evaluate the role of ^18^F-fluoro-L-dihydroxyphenylalanine ^(18^F-FDOPA) positron emission tomography (PET) in the context of glioblastoma multiforme, examining its clinical utility and potential benefits over existing imaging techniques. This includes a detailed analysis of its ability to enhance diagnostic precision, guide therapeutic decisions, and ultimately improve patient outcomes in GBM management.

## 2. Diagnosis of the Glioblastoma Multiforme

The diagnosis of GBM employs a multifaceted approach integrating clinical, radiographic, and histopathological techniques to achieve accurate characterization and staging of the tumor [1,2,3,4,5]. 

### 2.1. The Use of Computed Tomography (CT) Imaging

Computed Tomography (CT) imaging plays a crucial role in the diagnosis and management of glioblastoma, a highly aggressive primary brain tumor. While MRI is the gold standard for brain tumor imaging due to its superior soft tissue contrast, CT imaging is invaluable for its rapid acquisition, availability, and utility in specific clinical scenarios [1,2]. CT scans are often used in the initial evaluation of patients presenting with neurological symptoms to quickly identify mass effects, hemorrhage, and calcifications associated with glioblastoma. In the preoperative setting, CT imaging provides detailed information about the tumor’s relationship to the skull and the presence of bone involvement, which is vital for surgical planning [2,8]. Additionally, CT is used postoperatively to assess for complications such as hemorrhage, hydrocephalus, or residual tumor burden. In some cases, CT perfusion imaging can be employed to evaluate tumor vascularity, helping to differentiate between a tumor recurrence and treatment-related changes such as radiation necrosis. Despite its limitations in tissue characterization compared to MRI, CT imaging remains an essential tool in the comprehensive care of glioblastoma patients [1,2,8,9].

### 2.2. The Use of Magnetic Resonance Imaging (MRI)

MRI remains the cornerstone of radiographic diagnosis, with contrast-enhanced T1-weighted imaging providing critical information on the presence of the hallmark ring-enhancing lesion, which is indicative of a necrotic core surrounded by a highly vascularized periphery [2]. Advanced MRI techniques such as diffusion-weighted imaging (DWI), perfusion-weighted imaging (PWI), and magnetic resonance spectroscopy (MRS) further augment diagnostic precision by assessing tumor cellularity, vascularity, and the metabolic profile, respectively [11,12]. 

#### 2.2.1. Contrast-Enhanced T1-Weighted Imaging

Contrast-enhanced MRI is integral to the management of glioblastoma patients, providing critical insights at various stages of the disease. This imaging modality leverages gadolinium-based contrast agents to enhance tumor visualization, exploiting the disrupted blood–brain barrier characteristic of high-grade gliomas [1,2,5,6,7]. It enables the precise delineation of tumor boundaries, aiding in accurate diagnoses and facilitating the differentiation of glioblastoma from other brain pathologies. Moreover, contrast-enhanced MRI is indispensable in pre-surgical planning, guiding neurosurgeons by delineating tumor margins and adjacent eloquent brain regions. Postoperatively, it plays a vital role in monitoring the treatment response and detecting recurrence by distinguishing between treatment-related changes and tumor progression [13].

#### 2.2.2. Diffusion-Weighted Imaging (DWI)

Diffusion-weighted imaging (DWI) is a specialized MRI technique that measures the random Brownian motion of water molecules within tissues. In the context of glioblastoma multiforme (GBM), DWI provides valuable insights into tumor cellularity and the integrity of the cellular environment [11]. GBM typically presents as areas of restricted diffusion due to its high cellular density, which impedes water molecule’s movement. This restriction is quantified by the apparent diffusion coefficient (ADC), where lower ADC values correlate with higher tumor cellularity and aggressiveness [11,12]. DWI is particularly useful in differentiating GBM from other types of brain lesions and in assessing tumor response to therapy by monitoring changes in cellularity over time. Additionally, DWI can help distinguish between recurrent tumors and treatment-induced changes such as radiation necrosis, where the latter typically shows higher ADC values due to reduced cellular density [11,12,14].

#### 2.2.3. Perfusion-Weighted Imaging (PWI)

Perfusion-weighted imaging (PWI) evaluates the hemodynamic properties of brain tissues by measuring the cerebral blood volume (CBV), cerebral blood flow (CBF), and mean transit time (MTT). In GBM, PWI is used to assess the tumor’s vascular characteristics, which are often markedly abnormal due to the presence of neoangiogenesis [15]. High-grade gliomas like GBM exhibit elevated CBV and CBF in the tumor core and periphery compared to normal brain tissue, reflecting their high vascularity and metabolic demands. These perfusion metrics are crucial for tumor grading, with higher values indicating more aggressive tumor behavior. PWI is also instrumental in treatment planning and monitoring, as changes in perfusion parameters can indicate a therapeutic response or tumor progression [15,16]. For instance, a decrease in CBV post-treatment may suggest a positive response, while persistently high or increasing CBV may signal recurrence or treatment resistance [15,16,17].

#### 2.2.4. Functional Magnetic Resonance Imaging (fMRI)

Functional Magnetic Resonance Imaging (fMRI) is a pivotal tool in the management of glioblastoma patients, offering crucial functional insights that complement anatomical imaging. By measuring blood oxygen level-dependent (BOLD) changes, fMRI maps brain activity associated with motor, sensory, and language functions [18]. This capability is especially valuable in preoperative planning, as it helps neurosurgeons identify and preserve critical functional areas adjacent to or within the tumor [19]. Additionally, fMRI aids in assessing the functional integrity of the brain, guiding the development of personalized therapeutic approaches that minimize cognitive and neurological deficits. In the context of glioblastoma, where tumors often infiltrate eloquent brain regions, fMRI’s ability to delineate functional networks is indispensable for optimizing surgical resection while preserving quality of life. Thus, fMRI not only enhances the precision of surgical interventions, but it also contributes to the broader therapeutic and prognostic landscape in glioblastoma care [18,19,20].

#### 2.2.5. Magnetic Resonance Spectroscopy (MRS)

Magnetic resonance spectroscopy (MRS) provides a non-invasive method to analyze the chemical composition of brain tissues [21]. In GBM, MRS detects alterations in the concentrations of various metabolites, offering insights into the tumor’s metabolic profile. Key metabolites analyzed include choline (Cho), creatine (Cr), N-acetylaspartate (NAA), lactate, and myo-inositol. GBM is typically characterized by elevated Cho levels, reflecting increased membrane turnover and cellular proliferation, and decreased NAA levels, indicative of neuronal loss or dysfunction [21,22]. The Cho/NAA ratio is particularly useful in differentiating high-grade gliomas from lower-grade tumors and non-neoplastic lesions. Elevated lactate and lipid peaks may also be observed, correlating with anaerobic metabolism and necrosis, respectively. MRS aids in tumor grading, guiding biopsy, and distinguishing tumor recurrence from treatment-related changes by identifying distinct metabolic patterns associated with each condition [23].

#### 2.2.6. CEST (Chemical Exchange Saturation Transfer) MRI

Chemical Exchange Saturation Transfer (CEST) MRI is an advanced imaging technique that exploits the exchange of protons between endogenous metabolites or exogenous contrast agents and bulk water to generate contrast [24]. This method involves selectively saturating the magnetization of exchangeable protons on metabolites or contrast agents using radiofrequency pulses. These saturated protons then exchange with the bulk water protons, leading to a reduction in the water signal that can be detected and quantified. CEST MRI provides a unique contrast mechanism sensitive to specific molecular environments and concentrations, making it useful for studying metabolic changes, pH variations, and protein interactions within tissues [24,25]. The technique’s high sensitivity to chemical exchange processes enables detailed molecular imaging that can enhance the understanding of various physiological and pathological states, offering potential applications in oncology, neurology, and cardiology [24,25,26].

### 2.3. Biopsy

#### 2.3.1. Stereotactic Biopsy

Stereotactic biopsy plays a crucial role in the management of glioblastoma patients, offering a minimally invasive method to obtain tissue samples for definitive histopathological diagnosis [2]. This technique utilizes three-dimensional imaging guidance, such as MRI or CT, to precisely target and sample tumor regions, ensuring an accurate diagnosis while minimizing the surgical risks. Stereotactic biopsy is particularly valuable in cases where the tumor is located in eloquent or deep-seated brain areas, where conventional surgical approaches may pose significant risks. Additionally, it allows for the molecular and genetic profiling of glioblastomas, which is essential for personalized treatment planning and prognostic assessments. By providing critical diagnostic and molecular information with minimal morbidity, stereotactic biopsy enhances the overall management and therapeutic strategy for glioblastoma patients [27,28].

#### 2.3.2. Open Biopsy

Open biopsy plays a pivotal role in the management of glioblastoma patients, serving as a crucial diagnostic procedure when non-invasive imaging and less invasive biopsy techniques are inconclusive or insufficient. This surgical approach involves the direct visualization and resection of tumor tissue, allowing for a comprehensive histopathological examination and an accurate diagnosis [28]. Open biopsy is particularly advantageous in providing ample tissue samples for detailed molecular and genetic analyses, which are essential for the characterization of tumor heterogeneity and the identification of therapeutic targets. Moreover, it enables the assessment of the tumor’s spatial context within the brain, facilitating more informed surgical planning and subsequent interventions. Despite its invasiveness compared to stereotactic biopsy, open biopsy remains a valuable procedure in glioblastoma management, ensuring diagnostic precision and supporting the development of personalized treatment strategies [27,28,29,30].

### 2.4. Histopathological Analysis

#### 2.4.1. Immunochemistry

Immunohistochemistry (IHC) plays a critical role in the management of glioblastoma patients by providing detailed insights into the tumor’s molecular and cellular characteristics [31]. This technique utilizes specific antibodies to detect and visualize the expression of proteins associated with glioblastoma, such as GFAP (glial fibrillary acidic protein), Ki-67, and IDH1 (isocitrate dehydrogenase 1) [32]. The expression profiles obtained through IHC are essential for confirming the diagnosis and distinguishing glioblastomas from other central nervous system tumors. Additionally, IHC is instrumental in identifying prognostic markers and potential therapeutic targets, thereby guiding personalized treatment strategies. For instance, the presence of IDH1 mutations, detected via IHC, is associated with a better prognosis and can influence therapeutic decisions [33]. By providing a precise molecular characterization of glioblastomas, immunohistochemistry enhances diagnostic accuracy, informs the prognosis, and aids in the selection of targeted therapies, ultimately contributing to more effective and individualized patient care [31,32,33,34].

#### 2.4.2. Molecular and Genetic Testing

Molecular and genetic testing are indispensable tools in the management of glioblastomas, offering critical insights that influence diagnosis, prognosis, and treatment strategies. These tests analyze tumor DNA and RNA to identify key genetic alterations, such as mutations in the IDH1 and IDH2 genes, MGMT promoter methylation, and EGFR amplification, which are pivotal in glioblastoma classification and prognosis. The identification of IDH mutations, for example, is associated with a better prognosis and can influence therapeutic approaches [35]. The MGMT promoter methylation status is another crucial biomarker that predicts responsiveness to alkylating agents like temozolomide, thereby guiding chemotherapy decisions. Furthermore, comprehensive genetic profiling enables the identification of novel therapeutic targets and the development of personalized treatment regimens, including targeted therapies and immunotherapies [35,36].

### 2.5. The Use of Positron Emission Imaging (PET)

#### 2.5.1. ^18^F-DG-PET

^18^F-fluorodeoxyglucose positron emission tomography (^18^F-FDG PET) is a vital imaging modality in the management of glioblastoma patients, providing metabolic insights that complement anatomical and functional imaging techniques [37]. ^18^F-FDG PET leverages the increased glucose metabolism characteristic of malignant cells to identify hypermetabolic regions within the brain, thus aiding in the differentiation of glioblastomas from other brain lesions and benign conditions. This modality is particularly useful in assessing the tumor grade and heterogeneity, as a higher metabolic activity often correlates with more aggressive tumor behavior [37,38]. Additionally, ^18^F-FDG PET is instrumental in distinguishing between a tumor recurrence and treatment-related changes, such as radiation necrosis, which can present similarly on conventional imaging [39].

#### 2.5.2. Amino Acid PET

PET with radiolabeled amino acids (e.g., FDOPA-PET, O-(2-[^18^F]fluoroethyl)-L-tyrosine (FET-PET), L-[methyl-11C]methionine (MET-PET)) offers complementary metabolic imaging that can differentiate a tumor recurrence from treatment-induced changes, providing valuable insights for clinical decision making [39]. In this article, we demonstrate the role of ^18^F-FDOPA PET in patients with GBM.

##### ^18^F-FDOPA PET

^18^F-FDOPA is a radiolabeled amino acid analog (Figure 1.) used in PET imaging, providing valuable insights into the metabolic activity of brain tumors, including glioblastomas. ^18^F-FDOPA is taken up by amino acid transporters that are typically overexpressed in tumor cells, allowing for the visualization and quantification of the tumor metabolism [40]. This imaging modality is particularly effective in differentiating a tumor recurrence from treatment-related changes such as radiation necrosis, a common challenge in the post-therapeutic assessment of glioblastoma patients. The enhanced uptake of ^18^F-FDOPA in glioblastomas, compared to normal brain tissue, facilitates accurate tumor delineation, an assessment of tumor aggressiveness, and the monitoring of the treatment response [41]. Furthermore, ^18^F-FDOPA PET imaging contributes to the non-invasive evaluation of tumor biology, aiding in the selection and evaluation of targeted therapies. By providing precise metabolic and functional information, ^18^F-FDOPA PET enhances the diagnostic and therapeutic management of glioblastoma patients [40,41,42,43,44].

##### Synthetization of ^18^F-FDOPA PET

The synthesis of ^18^F-FDOPA for PET imaging involves a multi-step chemical process that begins with the production of the radioactive isotope ^18^F. This is typically generated via proton bombardment of ^18^O-enriched water in a cyclotron, producing ^18^F-fluoride [40]. The ^18^F-fluoride is then reacted with a suitable precursor molecule through nucleophilic substitution to introduce the fluorine-18 atom into the DOPA structure [40,41]. The synthesis proceeds with the protection of the hydroxyl groups and the amino group on the DOPA molecule to prevent unwanted side reactions. After the ^18^F-labeling step, these protecting groups are removed in a deprotection step to yield the final ^18^F-FDOPA product [45]. The final compound is purified using high-performance liquid chromatography (HPLC) to ensure radiochemical purity and the specific activity suitable for clinical use. This synthesis process, while complex, results in a radiotracer that can be used to evaluate amino acid transport and metabolism in brain tumors [45,46].

##### Imaging Protocol of ^18^F-FDOPA PET

Before imaging, patients are typically required to fast for at least 4–6 h to minimize interference from dietary amino acids, which can affect ^18^F-FDOPA uptake. The injection dose of ^18^F-FDOPA is generally calculated based on body weight, with a standard dose being around 185–370 MBq (5–10 mCi). After the tracer injection, an uptake period of approximately 60 min is recommended to allow for sufficient tracer distribution and optimal imaging contrast between the tumor and the surrounding brain tissue [43,44]. 

During the scan, the imaging duration usually ranges from 20 to 30 min, with the scanner covering the region from the skull base to the vertex to ensure comprehensive brain coverage. It is important to ensure that the patient remains still during this period to avoid motion artifacts, which can compromise image quality. The acquired images are then typically reconstructed using algorithms that account for attenuation correction and scatter, enhancing the accuracy of the data [44]. This protocol should also include specific instructions for managing potential artifacts and a standardized approach to image interpretation to distinguish between a tumor recurrence and treatment effects. By adhering to these detailed protocols, ^18^F-FDOPA PET imaging can provide more reliable and reproducible results, improving its diagnostic utility in neuro-oncology [43,44].

##### Diagnostic Accuracy of ^18^F-FDOPA PET in Detecting Glioblastoma

The diagnostic accuracy of ^18^F-FDOPA PET has been extensively evaluated for detecting GBM, demonstrating notable sensitivity and specificity. ^18^F-FDOPA PET capitalizes on the elevated amino acid transport in tumor cells, providing high-contrast images that distinguish tumor tissue from the surrounding normal brain parenchyma [40,41,42,43,44]. Studies have reported sensitivity rates for ^18^F-FDOPA PET in GBM detection ranging from 80% to 90%, with specificity rates approximately ranging from 70% to 85% [42,43,44]. These metrics surpass those of conventional imaging modalities such as CT, which has limited utility in brain tumor imaging due to poor soft tissue contrast, and are comparable to or even exceed the performance of standard MRI. While MRI remains the primary imaging modality for initial diagnosis and surgical planning in GBM due to its superior anatomical resolution and ability to detect peritumoral edema and necrosis, ^18^F-FDOPA PET provides crucial metabolic information that complements MRI findings [42]. Furthermore, ^18^F-FDOPA PET exhibits distinct advantages over MRI in distinguishing tumor recurrence from radiation necrosis, a common challenge in post-treatment evaluation. When integrated with advanced MRI techniques like perfusion-weighted imaging (PWI) and MRS, ^18^F-FDOPA PET significantly enhances diagnostic accuracy, offering a comprehensive approach that improves the detection, characterization, and management of GBM [47,48].

##### The Role of ^18^F-FDOPA PET in Defining Target Volumes

^18^F-FDOPA PET plays a crucial role in defining target volumes for radiotherapy in glioblastoma patients by providing precise metabolic imaging that complements anatomical MRI [42]. This advanced imaging technique highlights areas of increased amino acid transport and metabolic activity, which are indicative of viable tumor tissue. By accurately delineating the metabolically active tumor regions, ^18^F-FDOPA PET enables radiation oncologists to better define the gross tumor volume (GTV) and the clinical target volume (CTV), ensuring that the most aggressive tumor areas are adequately covered while sparing healthy brain tissue [42]. This precision is particularly important in glioblastomas, where tumor infiltration beyond the visible margins on conventional MRI is common. Additionally, ^18^F-FDOPA PET can help identify regions of subclinical disease and distinguish between a tumor recurrence and treatment-induced changes, such as radiation necrosis. Therefore, the integration of ^18^F-FDOPA PET into radiotherapy planning enhances the accuracy of target volume delineation, potentially improving treatment efficacy and reducing radiation-induced toxicity [42,49,50,51].

##### Radiotherapy Response Monitoring with ^18^F-FDOPA PET

The efficacy of ^18^F-FDOPA PET in monitoring tumor response to radiotherapy in GBM patients has shown promising results, particularly in distinguishing between a tumor recurrence and radiation necrosis. Specifically, ^18^F-FDOPA PET has been shown to have a sensitivity ranging from 81% to 96%, and a specificity ranging from 77% to 100% in distinguishing between a tumor recurrence and radiation necrosis **[42,52]**. Post-radiotherapy, conventional imaging modalities like MRI often face challenges in differentiating treatment-related changes from tumor regrowth due to overlapping radiographic features. ^18^F-FDOPA PET addresses this limitation by exploiting the differential metabolic activity between viable tumor cells and necrotic tissue. Tumor cells exhibit increased amino acid transport and the uptake of ^18^F-FDOPA, resulting in higher PET signal intensities; whereas, radiation necrosis, characterized by non-viable tissue, shows reduced or absent tracer uptake [52,53]. Studies have demonstrated that ^18^F-FDOPA PET can reliably detect metabolic changes in GBM with a sensitivity and specificity that often surpass those of MRI alone. Furthermore, a quantitative analysis of ^18^F-FDOPA uptake allows for the assessment of metabolic response to radiotherapy over time, providing valuable prognostic information and aiding in the optimization of treatment plans (Figure 2) [52,53,54,55].

##### Biomarker Potential of ^18^F-FDOPA PET in Glioblastoma

The biomarker potential of ^18^F-FDOPA PET in GBM has gained attention due to its correlation with molecular markers and prognostic significance. Studies have demonstrated a significant association between ^18^F-FDOPA uptake and key molecular features of GBM, such as isocitrate dehydrogenase (IDH) mutations, the O6-methylguanine-DNA methyltransferase (MGMT) promoter methylation status, and epidermal growth factor receptor (EGFR) amplification [56,57]. A high ^18^F-FDOPA uptake often correlates with more aggressive tumor phenotypes and poorer prognostic markers, providing insights into tumor biology and potential therapeutic targets. Additionally, quantitative metrics derived from ^18^F-FDOPA PET, such as the maximum standardized uptake value (SUVmax) and tumor-to-normal brain tissue ratio (TBR), have been shown to predict patient outcomes. An elevated ^18^F-FDOPA uptake is associated with shorter progression-free survival (PFS) and overall survival (OS), underscoring its prognostic value [56,57,58]. As a non-invasive imaging biomarker, ^18^F-FDOPA PET facilitates the monitoring of tumor behavior and the treatment response, enabling more precise and personalized therapeutic strategies [59].

##### Clinical Workflow Integration of ^18^F-FDOPA PET

Incorporating ^18^F-FDOPA PET into routine clinical practice for GBM involves addressing several practical and logistical considerations to ensure its effective utilization. The standardization of imaging protocols is paramount to achieving consistent and reliable results across different clinical settings [42]. This includes uniform guidelines for patient preparation, tracer administration, imaging acquisition parameters, and data interpretation. Optimal timing for ^18^F-FDOPA PET scans relative to other treatments, such as surgery, radiotherapy, and chemotherapy, must be established to maximize diagnostic accuracy and clinical relevance [40,41,42,43,44]. Integrating ^18^F-FDOPA PET into the existing clinical workflow also requires collaboration among multidisciplinary teams, including nuclear medicine specialists, radiologists, oncologists, and neurosurgeons, to interpret findings and incorporate them into treatment planning [47,48]. Furthermore, ensuring accessibility to ^18^F-FDOPA PET imaging involves logistical arrangements for tracer production, distribution, and scheduling within the healthcare facility. Developing standardized reporting frameworks and incorporating PET findings into electronic health records (EHRs) can facilitate the seamless communication and longitudinal tracking of patient outcomes [60].

##### Impact of ^18^F-FDOPA PET-Guided Management on Patient Outcomes

The integration of ^18^F-FDOPA PET into the management of GBM has shown promising impacts on patient survival and quality of life [60]. Clinical trials and case studies have demonstrated that ^18^F-FDOPA PET-guided management can lead to more accurate differentiation between a tumor recurrence and treatment-related changes such as radiation necrosis, facilitating timely and appropriate therapeutic interventions. This enhanced diagnostic precision enables more tailored treatment regimens, potentially avoiding unnecessary or ineffective therapies and reducing adverse effects. For instance, patients monitored with ^18^F-FDOPA PET can benefit from early identification of tumor progression, allowing for prompt adjustments in treatment strategies, such as the introduction of alternative chemotherapeutic agents or the implementation of additional surgical interventions [61,62].

Case studies have illustrated improved progression-free survival (PFS) and overall survival (OS) in patients whose treatment plans were guided by ^18^F-FDOPA PET findings [63]. Moreover, the ability to non-invasively monitor the metabolic activity and therapeutic response contributes to better-informed clinical decisions, ultimately enhancing patient care. In terms of quality of life, ^18^F-FDOPA PET helps in minimizing the physical and psychological burdens associated with misdiagnosis and overtreatment, promoting a more patient-centered approach to GBM management. Overall, the incorporation of ^18^F-FDOPA PET into clinical practice not only optimizes the therapeutic efficacy, but also supports improved survival outcomes and quality of life for GBM patients, as evidenced by accumulating clinical data [60,61,62].

##### Technological and Methodological Considerations in ^18^F-FDOPA PET Imaging

Advancements in ^18^F-FDOPA PET imaging technology have significantly enhanced its application in the diagnosis and management of GBM. Recent technological developments include the improvement of PET scanner resolution and sensitivity, which enable a more precise detection of metabolic activity within brain tumors. High-resolution PET/CT and PET/MRI hybrid systems combine anatomical and functional imaging, providing comprehensive insights into tumor characteristics and the surrounding brain structures. These integrated imaging modalities facilitate more accurate localization and delineation of GBM, aiding in better surgical planning and treatment monitoring [64].

The optimization of imaging techniques and interpretation criteria is crucial for maximizing the diagnostic and prognostic utility of ^18^F-FDOPA PET. Standardizing imaging protocols, such as tracer dosage, the timing of image acquisition, and patient preparation, ensures consistency and reproducibility across clinical settings. Quantitative analysis methods, including the calculation of standardized uptake values (SUVs) and tumor-to-normal brain tissue ratios (TBRs), are essential for an objective assessment of ^18^F-FDOPA uptake [42,53]. Advanced image processing algorithms and artificial intelligence (AI) applications are being developed to automate and enhance the accuracy of these quantitative measures, reducing inter-observer variability and improving diagnostic confidence [60].

Interpretation criteria are also evolving to incorporate multi-parametric data, integrating metabolic, anatomical, and molecular information to provide a more comprehensive evaluation of the tumor status. Consensus guidelines and training programs for clinicians and radiologists are necessary to ensure the effective implementation and interpretation of ^18^F-FDOPA PET imaging in clinical practice. As these technological and methodological advancements continue to evolve, they hold the potential to further refine the role of ^18^F-FDOPA PET in GBM management, ultimately improving patient outcomes through more precise and personalized therapeutic approaches [60,65].

##### Comparative Studies of ^18^F-FDOPA PET versus Other PET Tracers in Glioblastoma

Direct comparison studies and meta-analyses have been pivotal in evaluating the relative efficacy of ^18^F-FDOPA PET against other PET tracers used in glioblastoma multiforme (GBM) imaging (Table 1.). Commonly compared tracers include ^18^F-fluorodeoxyglucose (^18^F-FDG), ^11^C-methionine (^11^C-MET) [39], and ^18^F-fluorothymidine (^18^F-FLT) [66], each offering unique insights into tumor metabolism and proliferation. ^18^F-FDOPA PET has shown superior specificity in distinguishing tumor tissue from normal brain parenchyma due to its lower background uptake in non-tumorous brain regions; a distinct advantage over ^18^F-FDG PET, which often has a high physiological uptake in the brain, complicating interpretation [40,67].

^11^C-MET PET, another amino acid-based tracer, provides comparable sensitivity to ^18^F-FDOPA PET; however, ^18^F-FDOPA benefits from the longer half-life of fluorine-18, making it more practical for widespread clinical use due to logistical ease in synthesis and distribution [68]. Studies have indicated that ^18^F-FDOPA PET is particularly effective in identifying recurrent GBM and differentiating it from post-treatment changes, a challenge where ^18^F-FLT PET has shown limitations despite its specificity in highlighting proliferative activity. Specifically, ^18^F-FDOPA PET has been reported to have a sensitivity of approximately 83% to 100% and a specificity of around 50% to 95% in detecting recurrent GBM [68,69,70].

Meta-analyses of existing research data corroborate these findings, consistently showing that ^18^F-FDOPA PET offers a robust combination of high sensitivity and specificity for GBM imaging, outperforming other tracers in several critical diagnostic areas. These analyses aggregate data across multiple studies, reinforcing the reliability of ^18^F-FDOPA PET in clinical applications and highlighting its role in improving diagnostic accuracy, treatment planning, and patient outcomes. As research continues, further comparative studies and meta-analyses will be essential in refining the clinical protocols and expanding the adoption of ^18^F-FDOPA PET in the management of glioblastoma [70,71].

**Table 1 pharmaceuticals-17-01215-t001:** The advantages and disadvantages of ^18^F-FDOPA, ^18^F-FDG, ^11^C-MET, and ^18^F-FET in the imaging of glioblastoma [37,38,39,40,42,49,68,70,71].

Tracer	Advantages	Disadvantages
^18^F-FDOPA	- High sensitivity and specificity for detecting glioblastoma and distinguishing it from treatment-related changes. - Longer half-life of fluorine-18 (110 min) allows for wider distribution and use in clinical settings. - Good correlation with the tumor grade and potential for prognostic evaluation.	- Uptake can sometimes occur in inflammatory tissues, potentially leading to false positives. - Limited availability compared to other tracers like ^18^F-FDG.
^18^F-FDG	- Widely available and commonly used PET tracer. - Provides valuable information on glucose metabolism, which is often increased in high-grade tumors like GBM.	- Poor differentiation between tumor and normal brain tissue due to the high background uptake of glucose in the brain. - Less effective in distinguishing tumor recurrence from radiation necrosis.
^11^C-MET	- High sensitivity for detecting tumor activity and monitoring recurrence. - Better contrast between tumor and normal brain tissue compared to ^18^F-FDG. - Shorter scan times due to rapid uptake and clearance.	- Short half-life of carbon-11 (20 min) limits its use to facilities with an on-site cyclotron. - Limited availability and higher logistical complexity.
^18^F-FET	- High specificity for glioma tissue with low uptake in inflammatory tissue, reducing false positives. - Useful for tumor grading and assessing the treatment response. - Longer half-life of fluorine-18 allows for wider clinical use.	- Lower sensitivity compared to ^11^C-MET in some cases. - Limited differentiation between low-grade and high-grade gliomas.

##### Cost-Effectiveness of ^18^F-FDOPA PET in Glioblastoma Treatment

The economic evaluation of ^18^F-FDOPA PET in the treatment of GBM is an essential consideration given the high costs associated with advanced imaging technologies. Cost-effectiveness analyses have demonstrated that ^18^F-FDOPA PET can provide substantial clinical benefits that justify its expenses, particularly through improved diagnostic accuracy and optimized treatment planning. By accurately distinguishing between a tumor recurrence and treatment-related changes, ^18^F-FDOPA PET reduces the likelihood of unnecessary and costly interventions, such as repeat surgeries and inappropriate chemoradiotherapy regimens. This precision not only decreases direct medical costs, but also minimizes patient morbidity associated with incorrect treatment approaches, potentially lowering hospitalization and rehabilitation expenses [72,73].

Moreover, ^18^F-FDOPA PET’s ability to monitor the treatment response more effectively than conventional imaging modalities can lead to earlier adjustments in therapeutic strategies, enhancing the overall treatment efficacy and potentially prolonging progression-free survival (PFS) and overall survival (OS). The long-term economic benefits of improved patient outcomes include increased productivity and reduced indirect costs associated with caregiving and the loss of work. Several studies have quantified these benefits, demonstrating that the initial higher cost of ^18^F-FDOPA PET is offset by the savings accrued from more effective and efficient clinical management [73].

In comprehensive economic evaluations, the incremental cost-effectiveness ratio (ICER) of ^18^F-FDOPA PET has been found to be within acceptable thresholds when compared to traditional imaging techniques. This is especially relevant when considering the broader healthcare system and societal perspectives, where the value of advanced imaging in enhancing quality-adjusted life years (QALYs) is a critical metric. As the healthcare landscape continues to evolve with a focus on precision medicine, the integration of ^18^F-FDOPA PET into routine GBM management is increasingly recognized not only for its clinical advantages, but also for its favorable cost-effectiveness profile [74].

## 3. Discussion

The integration of ^18^F-FDOPA PET into the management of GBM represents a significant advancement in neuro-oncology, offering enhanced diagnostic precision and therapeutic guidance. This imaging modality capitalizes on the elevated amino acid transport in tumor cells, providing high-contrast images that effectively distinguish tumor tissue from surrounding normal brain parenchyma. The ability of ^18^F-FDOPA PET to accurately delineate metabolically active tumor regions facilitates more precise targeting for radiotherapy, ensuring a comprehensive coverage of the gross tumor volume (GTV) and clinical target volume (CTV) while sparing healthy tissue. This precision is particularly crucial in GBM, where the tumor’s infiltrative nature often extends beyond the visible margins on conventional MRI [1,2,40,41,42,43,44,45,46].

Furthermore, ^18^F-FDOPA PET plays a pivotal role in differentiating a tumor recurrence from treatment-related changes such as radiation necrosis. This distinction is critical for appropriate clinical decision making, as conventional imaging techniques like MRI often struggle to differentiate between these conditions due to overlapping radiographic features. The metabolic insights provided by ^18^F-FDOPA PET thus enable a more accurate assessment of therapeutic response and timely adjustments to treatment plans, potentially improving patient outcomes [44,47,51,55].

From a molecular perspective, ^18^F-FDOPA PET’s ability to correlate with key genetic alterations and prognostic markers in GBM, such as IDH mutations, MGMT promoter methylation, and EGFR amplification, underscores its potential as a non-invasive biomarker. This capability facilitates personalized treatment strategies, aligning therapeutic approaches with the molecular and metabolic profiles of individual tumors [46,59,62,63,65].

Despite the clinical benefits, the widespread adoption of ^18^F-FDOPA PET in routine practice faces practical and logistical challenges, including the need for standardized imaging protocols and multidisciplinary collaboration for optimal integration into clinical workflows. Moreover, cost-effectiveness analyses indicate that while ^18^F-FDOPA PET incurs higher initial costs, these are offset by the reduction in unnecessary interventions and improved patient management, ultimately proving its economic viability [61,62,69].

## 4. Limitations

Despite its numerous advantages, ^18^F-FDOPA PET has several limitations that can impact its clinical utility in GBM management. One significant limitation is the relatively high cost and limited availability of ^18^F-FDOPA PET, which can restrict access to this advanced imaging modality in many healthcare settings. Additionally, the interpretation of ^18^F-FDOPA PET images can be complex, requiring specialized expertise and standardized protocols to ensure accuracy and reproducibility. Another limitation is the potential for false positives and negatives; for instance, inflammation and other non-neoplastic processes can exhibit an increased ^18^F-FDOPA uptake, potentially confounding the differentiation between a tumor recurrence and treatment-induced changes. Furthermore, while ^18^F-FDOPA PET provides valuable metabolic information, it may still need to be combined with other imaging modalities, such as MRI, to achieve comprehensive diagnostic accuracy. The relatively short half-life of ^18^F also necessitates a nearby cyclotron facility for tracer production, which can pose logistical challenges. These limitations underscore the need for ongoing research to optimize ^18^F-FDOPA PET imaging protocols and integration into multimodal imaging strategies to fully realize its potential in GBM management.

## 5. Conclusions

Our review provides a novel and comprehensive analysis of the diagnostic value of ^18^F-FDOPA PET imaging in patients with GBM, highlighting its superiority in enhancing diagnostic accuracy, guiding therapeutic decisions, and improving patient outcomes. The integration of ^18^F-FDOPA PET with conventional imaging modalities, such as MRI and CT, as well as its correlation with histopathological findings, offers a more precise delineation of tumor margins and better differentiation between a tumor recurrence and treatment-related changes, such as radiation necrosis. This study’s findings underscore the potential of ^18^F-FDOPA PET as a crucial tool in the personalized management of GBM, providing both metabolic and molecular insights that are not achievable with traditional imaging techniques alone. 

The novelty of our review lies in its detailed evaluation of ^18^F-FDOPA PET’s role across different stages of GBM management, from diagnosis to treatment monitoring, and its comparison with other PET tracers, such as ^11^C-MET and ^18^F-FDG. This research contributes significantly to the evolving landscape of neuro-oncology by offering evidence-based recommendations for the incorporation of ^18^F-FDOPA PET into routine clinical practice, ultimately aiming to enhance both clinical and economic outcomes for patients with this formidable disease.

## Figures and Tables

**Figure 1 pharmaceuticals-17-01215-f001:**
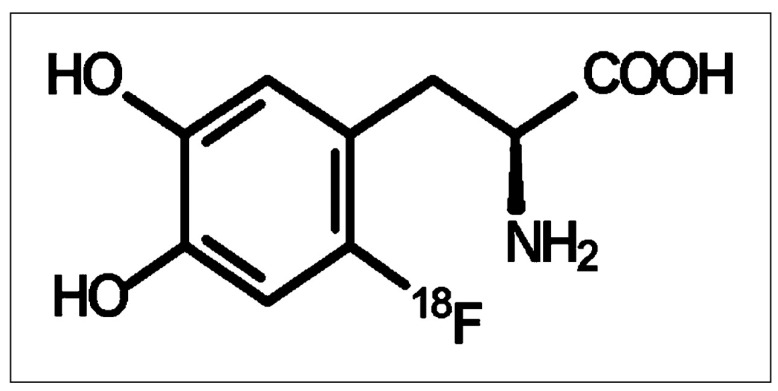
The chemical structure of ^18^F-FDOPA.

**Figure 2 pharmaceuticals-17-01215-f002:**
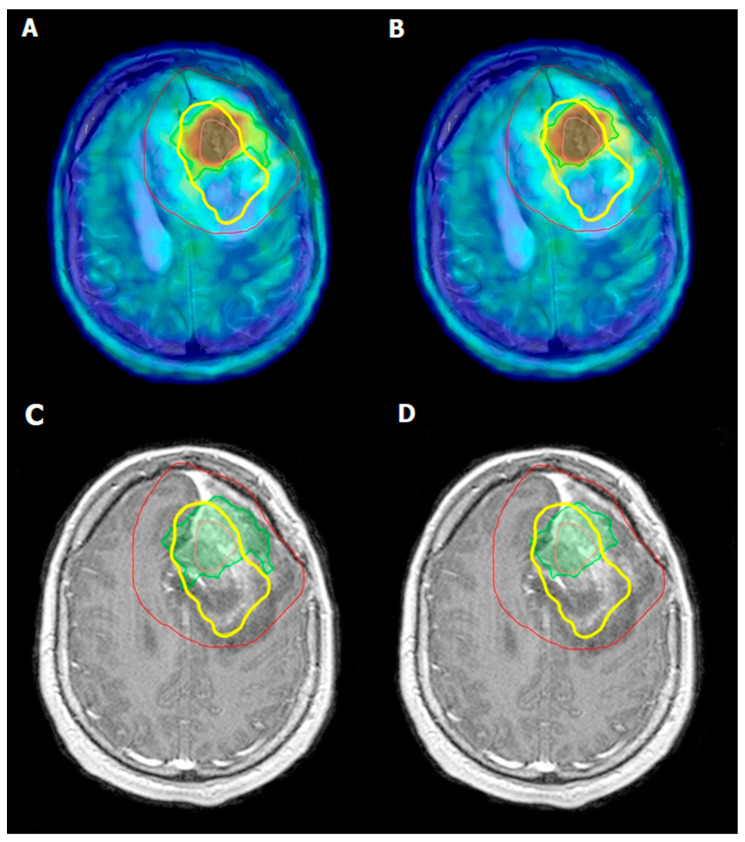
Left frontal post craniotomy status. Inhomogeneous, mainly centrally, moderate enhancement of contrast material is observed on T1-weighted post-contrast MRI images. The lesion in the left hemisphere is surrounded by edema (**C**,**D**). Irregularly shaped intense, focal ^18^F-FDOPA accumulation can be detected on the left side of the brain frontally, above the level of lateral ventricles (**A**,**B**). Pink line—GTV, green line—BTV 1.7 (**A**,**C**), green line—BTV 2.0 (**B**,**D**), red line—PTV, and yellow line—recurrence [42].

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
