# Peer review of "Complex Diagnostic Challenges in Glioblastoma: The Role of ^18^F-FDOPA PET Imaging"

_pharmaceuticals, 2024, doi:10.3390/ph17091215_

Round 1

Reviewer 1 Report

Comments and Suggestions for Authors

Thanks for this review, which will be of interest to readers engaged in radiology and brain imaging. 
First and foremost, the title needs revision as the word "patients" is redundant - remove one of them. I also recommend replacing "diagnostic value" with "role", as the content not only discusses the "diagnostic value" of the PET but also elaborates on its broader "role" in the managment of the Glioblastoma patinets. For example, the paper explains its role on monitoring patients after radiotherapy procedures.
There is also a comment that relates to the review of the state-of-the-art of technologies used for imaging and diagnosis of GBM. An important element is missing here: CT scan. This imaging modality is also considered and used as a tool in diagnosing Glioblastoma patients, even though it is less detailed than PET and MRI. It remains a useful tool for quickly assessing whether there is bleeding or other abnormalities in the brain. As your state-of-the-art tends to be comprehensive, it is expected to provide a complete picture of the tools that are used for diagnosing GBM. Therefore, it is recommended to add a section on CT to the "2. Diagnosis of the Glioblastoma Multiforme" to make this literature review of the imaging modalities comprehensive and complete.
The numbering of the sections needs revision. Both "Histopathological Analysis" and "The use of Positron Emission Imaging (PET)" have been numbered as 2.3, leading to some other repetitive numbering.
Uniformity of the terminology should also be controlled meticulously. For example, one of the 18FDG PET and 18FDG-PET should be revised to follow the style of other to keep uniformity in the text.
MRI is mentioned with full name and abbreviation several times. It should be introduced with its full name followed by the abb. when it first appear within the text, and thereafter only abbreviation should be used.
While MRI is fully named multiple times, the word "CT" has been used without its full name.
Overall the paper is written well but missing the CT section may be misleading. It should be added before the manuscript can be considered for the next step.

Author Response

Dear Reviewer 1!

Thank you sincerely for your thoughtful and constructive feedback on our work. Your insights have been invaluable in refining and improving the quality of our research. We appreciate your time and expertise. 

Regarding your comments:

Thanks for this review, which will be of interest to readers engaged in radiology and brain imaging.

  1. First and foremost, the title needs revision as the word "patients" is redundant - remove one of them. I also recommend replacing "diagnostic value" with "role", as the content not only discusses the "diagnostic value" of the PET but also elaborates on its broader "role" in the managment of the Glioblastoma patinets. For example, the paper explains its role on monitoring patients after radiotherapy procedures. – THANK YOU SO MUCH FOR YOUR FEEDBACK, WE’VE SUGGESTED A NEW TITLE
  2. There is also a comment that relates to the review of the state-of-the-art of technologies used for imaging and diagnosis of GBM. An important element is missing here: CT scan. This imaging modality is also considered and used as a tool in diagnosing Glioblastoma patients, even though it is less detailed than PET and MRI. It remains a useful tool for quickly assessing whether there is bleeding or other abnormalities in the brain. As your state-of-the-art tends to be comprehensive, it is expected to provide a complete picture of the tools that are used for diagnosing GBM. Therefore, it is recommended to add a section on CT to the "2. Diagnosis of the Glioblastoma Multiforme" to make this literature review of the imaging modalities comprehensive and complete. – THANK YOU, WE’VE ADDED THE CT SECTION AS WELL
  3. The numbering of the sections needs revision. Both "Histopathological Analysis" and "The use of Positron Emission Imaging (PET)" have been numbered as 2.3, leading to some other repetitive numbering.- CORRECTED
  4. Uniformity of the terminology should also be controlled meticulously. For example, one of the 18FDG PET and 18FDG-PET should be revised to follow the style of other to keep uniformity in the text.- CORRECTED
  5. MRI is mentioned with full name and abbreviation several times. It should be introduced with its full name followed by the abb. when it first appear within the text, and thereafter only abbreviation should be used.- CORRECTED
  6. While MRI is fully named multiple times, the word "CT" has been used without its full name. - CORRECTED
  7. Overall the paper is written well but missing the CT section may be misleading. It should be added before the manuscript can be considered for the next step. – WE’VE CORRECTED, THANK YOU VERY MUCH FOR YOUR TIME AND REVIEW ONCE AGAIN

Reviewer 2 Report

Comments and Suggestions for Authors

In the present review article, the authors provide an overview of the use of FDOPA PET imaging in glioblastoma patients. The manuscript is well written, showing the clinical use of FDOPA PET; not only discussing the role of FDOPA PET in diagnosis, radiotherapy planning, and detecting recurrence, but also addressing cost-effectiveness and limitation of the PET imaging.

However, there are several points that need to be clarified.

1. Please, us the superscript for “18” in the “18F” throughout the manuscript. Furthermore, for FDG, it might be better to write as “FDG” or “18F-FDG” rather than “18FDG”.

2. I recommend to provide figures that showed representative cases of FDOPA PET images in detecting glioblastoma and differentiating tumor recurrence from radiation necrosis.

3. Since many readers may be not familiar with FDOPA PET scan, it would be better to add a section that addresses detailed information regarding the imaging protocol of FDOPA PET, including preparations of patients before imaging, injection dose, uptake time, scanning time, and so on…

4. To enhance the readers’ understanding, I recommend to provide specific numbers for the sensitivity and specificity of FDOPA PET for differentiating glioblastoma from radiation necrosis (in section 5 Radiotherapy response monitoring with 18F-FDOPA PET) and in comparisons of diagnostic ability of FDOPA PET with other PET traces in section 10 (Comparative studies of 18F-FDOPA PET versus other PET traces in glioblastoma).

5. As this article is focused on the clinical value of FDOPA PET imaging, it would be better to shorten the section 2.1 regarding the use of MRI in glioblastoma. The sections from 2.1.1 to 2.1.6. are too long and wordy, and I recommend to coalesce sections 2.1.1 to 2.1.6. into one section.

6. Abbreviations should be spelled out when first used, such as 18F-FDOPA PET in page 2, FET-PET and MET-PET in page 5.

Author Response

Dear Reviewer 2!

We extend our sincere gratitude for your thorough review of our work. Your constructive feedback and valuable suggestions have significantly enhanced the quality of our research. We appreciate your time, expertise, and thoughtful contributions to the refinement of our manuscript.

Regarding your comments:

In the present review article, the authors provide an overview of the use of FDOPA PET imaging in glioblastoma patients. The manuscript is well written, showing the clinical use of FDOPA PET; not only discussing the role of FDOPA PET in diagnosis, radiotherapy planning, and detecting recurrence, but also addressing cost-effectiveness and limitation of the PET imaging.

However, there are several points that need to be clarified.

  1. Please, us the superscript for “18” in the “18F” throughout the manuscript. Furthermore, for FDG, it might be better to write as “FDG” or “18F-FDG” rather than “18FDG”. - CORRECTED
  2. I recommend to provide figures that showed representative cases of FDOPA PET images in detecting glioblastoma and differentiating tumor recurrence from radiation necrosis. – WE’VE ADDED ONE IMAGE FROM OUR PREVIOUS ARTICLE
  3. Since many readers may be not familiar with FDOPA PET scan, it would be better to add a section that addresses detailed information regarding the imaging protocol of FDOPA PET, including preparations of patients before imaging, injection dose, uptake time, scanning time, and so on… - CORRECTED
  4. To enhance the readers’ understanding, I recommend to provide specific numbers for the sensitivity and specificity of FDOPA PET for differentiating glioblastoma from radiation necrosis (in section 5 Radiotherapy response monitoring with 18F-FDOPA PET) and in comparisons of diagnostic ability of FDOPA PET with other PET traces in section 10 (Comparative studies of 18F-FDOPA PET versus other PET traces in glioblastoma). – THANK YOU SO MUCH, WE’VE ADDED A TABLE
  5. As this article is focused on the clinical value of FDOPA PET imaging, it would be better to shorten the section 2.1 regarding the use of MRI in glioblastoma. The sections from 2.1.1 to 2.1.6. are too long and wordy, and I recommend to coalesce sections 2.1.1 to 2.1.6. into one section. – REGARDING TO OTHER TWO REVIEWERS, WE’VE CHANGED THE TITLE OF THE ARTICLE, NOW IT SUITS INTO THE THEME, THANK YOU FOR YOUR FEEDBACK
  6. Abbreviations should be spelled out when first used, such as 18F-FDOPA PET in page 2, FET-PET and MET-PET in page 5. – THANK YOU, CORRECTED

Reviewer 3 Report

Comments and Suggestions for Authors

The authors presented the paper "Evaluating the Diagnostic Value of 18F-FDOPA PET Imaging in Patients with Glioblastoma Patients"

1) The title of your paper is "Evaluating the Diagnostic Value of 18F-FDOPA PET Imaging in Patients with Glioblastoma Patients". However, the paper discussed several topics (see lines 74-76). That is why it is misleading. Moreover, in the abstract you write about PET but where are MRI and other things?

2) In this imaging paper, there are no picture examples. I highly recommend inserting in Sections 2, 5-9 some images of glioblastoma detections by these methods. It will be better to do subsections instead of 14 sections.

3) There are two sections 2.3

For section, "The use of Positron Emission Imaging", it will be good to present pictures of 18F probes and explain in pictures the synthesis of 18F-FDOPA PET. Moreover, the principle of these probes in the organism will be good to present.

4) Section 10. It would be excellent to present a table with the advantages, disadvantages, and qualitative characteristics of PET probes for imaging of glioblastoma. Moreover, have you checked a similar method as SPECT for glioblastoma detection for comparison with your data?

5) The conclusion section is poor for the review paper. Future outlooks are required too. The novelty of the paper should be clearly mentioned in the Introduction and Conclusion sections.

line 222 18? 18 for F should be 18F

Comments on the Quality of English Language

Minor editing of English language required.

Author Response

Dear Reviewer 3,

Thank you immensely for your insightful and detailed review of our manuscript. Your thoughtful comments and suggestions have proven invaluable in strengthening the overall quality of our work. We are genuinely appreciative of your time and expertise in contributing to the refinement of our research.

Regarding your comments:

The authors presented the paper "Evaluating the Diagnostic Value of 18F-FDOPA PET Imaging in Patients with Glioblastoma Patients"

1) The title of your paper is "Evaluating the Diagnostic Value of 18F-FDOPA PET Imaging in Patients with Glioblastoma Patients". However, the paper discussed several topics (see lines 74-76). That is why it is misleading. Moreover, in the abstract you write about PET but where are MRI and other things? – WE’VE CHANGED THE TITLE AND CORRECTED THE ABSTRACT

2) In this imaging paper, there are no picture examples. I highly recommend inserting in Sections 2, 5-9 some images of glioblastoma detections by these methods. It will be better to do subsections instead of 14 sections. – WE’VE ADDED ONE FIGURE AND ONE COMPREHENSIVE TABLE FOR BETTER UNDERSTANDING

3) There are two sections 2.3 - CORRECTED

4) For section, "The use of Positron Emission Imaging", it will be good to present pictures of 18F probes and explain in pictures the synthesis of 18F-FDOPA PET. Moreover, the principle of these probes in the organism will be good to present. – WE’VE ADDED ONE IMAGE REGARDING TUMORE RECURRENCE

4) Section 10. It would be excellent to present a table with the advantages, disadvantages, and qualitative characteristics of PET probes for imaging of glioblastoma. Moreover, have you checked a similar method as SPECT for glioblastoma detection for comparison with your data? – WE’VE ADDED ONE COMPREHENSIVE TABLE FOR BETTER UNDERSTANDING

5) The conclusion section is poor for the review paper. Future outlooks are required too. The novelty of the paper should be clearly mentioned in the Introduction and Conclusion sections. – WE’VE REWROTE THE CONCLUSION SECTION

line 222 18? 18 for F should be 18F - CORRECTED

Round 2

Reviewer 2 Report

Comments and Suggestions for Authors

The authors have revised the manuscript properly, according to my comments. I have no further comments.

However, before considering acceptance in the journal, one minor correction would be needed. Although, the authors have added Figure 1, the Figure 1 is not referred in the main text of the manuscript. 

Author Response

Dear Reviewer 2!

We extend our sincere gratitude for your thorough review of our work once again.

Regarding your comments:

Comments and Suggestions for Authors

The authors have revised the manuscript properly, according to my comments. I have no further comments. – THANK YOU VERY MUCH, WE ARE PLEASED

However, before considering acceptance in the journal, one minor correction would be needed. Although, the authors have added Figure 1, the Figure 1 is not referred in the main text of the manuscript. – WE’VE MADE CORRECTION

Reviewer 3 Report

Comments and Suggestions for Authors

Thank you for the revised paper.

However, one figure is not enough for a review paper. I highly recommend presenting more figures, such as the chemical structure of probes in Chem Draw (for section Synthethization of 307 18F-FDOPA PET, btw why not synthesis?), some more PET examples, etc. for better paper understanding.

It will be better to do subsections instead of 14 sections.

Comments on the Quality of English Language

Minor editing of English language required.

Author Response

Dear Reviewer 3,

Thank you immensely for your insightful and detailed review of our manuscript once again.

Regarding your comments:

Thank you for the revised paper.

However, one figure is not enough for a review paper. I highly recommend presenting more figures, such as the chemical structure of probes in Chem Draw (for section Synthethization of 307 18F-FDOPA PET, btw why not synthesis?), some more PET examples, etc. for better paper understanding. – THANK YOU FOR YOUR COMMENT, WE’VE ADDED A FIGURE OF 18F-FDOPA

It will be better to do subsections instead of 14 sections. – THANK YOU FOR YOUR SUGGESTION, WE’VE TRIED TO REORGANIZE THE SECTIONS